# Emotional Eating and Perfectionism as Predictors of Symptoms of Binge Eating Disorder: The Role of Perfectionism as a Mediator between Emotional Eating and Body Mass Index

**DOI:** 10.3390/nu14163361

**Published:** 2022-08-16

**Authors:** Elena Bernabéu-Brotóns, Carlos Marchena-Giráldez

**Affiliations:** Education and Psychology School, Universidad Francisco de Vitoria, 28223 Madrid, Spain

**Keywords:** emotional eating, binge eating, perfectionism, body mass index, mediators

## Abstract

(1) Background: Perfectionism has been linked to eating disorders and might be a risk factor for the appearance of eating pathologies. The aims of this study are (a) to verify the relationship between perfectionism, emotional eating (EE), binge eating (BE), and body mass index (BMI); (b) to identify the variables that predict BE symptoms and BMI; (c) to study the role of perfectionism as a mediator between EE and BMI. (2) Methods: 312 adult participants answered a cross-sectional survey that included the Multidimensional Perfectionism Scale, the Emotional Eater Questionnaire (EEQ), the Binge Eating Scale (BES), and a sociodemographic questionnaire including BMI. (3) Results: The results suggest a direct correlation between EE, BE, and BMI, showing that EE is a powerful predictor of BE symptoms and BMI. Furthermore, two dimensions of perfectionism have a mediator role between EE and BMI, specifically doubts and actions and concern over mistakes: the presence of these two components of perfectionism reverses the relationship between EE and BMI. (4) Conclusions: These results have significant implications for the understanding of the two different (pathological) eating patterns: intake restriction and overeating and should be considered in intervention programs.

## 1. Introduction

Perfectionism has been described as an inclination to establish exceptionally high standards and engage in excessively critical self-evaluations [1]. Perfectionism differs from a healthy attitude of striving to improve oneself, and it often results in emotional distress and negative affects [2,3]. It has been associated in numerous studies with multiple psychological disorders including anxiety [4], mood disorders [5], and has long been associated to eating disorders [6,7]. Individuals with eating disorders have shown higher levels of perfectionism than healthy controls [8,9]. It has been suggested that perfectionism can become a predisposing personality trait. in developing or maintaining eating pathologies [10,11,12], and may constitute a risk factor, itself or in interaction with other elements, for the development and maintenance of eating disorders, particularly anorexia nervosa [10,11,13]. Patients with anorexia who have sustained weight restoration appear to still struggle with perfectionism [14,15].

Until the early 1990s, perfectionism was conceptualized as a unidimensional construct. Subsequently, assessment measures for perfectionism began to reflect multidimensional conceptions of perfectionism, leading to the development of new instruments including the Frost Multidimensional Perfectionism Scale (Frost MPS) [1], and the Hewitt and Flett Multidimensional Perfectionism Scale (Hewitt and Flett MPS) [16]. Both tools have been employed frequently, demonstrating optimal psychometric properties [17], and both address interpersonal aspects of perfectionism. These multidimensional perfectionism measures have allowed researchers to explore which aspects of perfectionism are most strongly associated with eating disorder behaviors and to investigate the mechanisms by which perfectionism influences eating pathologies.

Emotional eating (EE) is the propensity to eat automatically as a reaction to unpleasant feelings [18], and refers to consuming large amounts of food despite not feeling particularly hungry [19]. Specifically, this consumption pattern has been linked to general (non-clinical) population anxiety and depressive symptoms [20]. Additionally, this form of eating habit exhibits a high rate of comorbidity in the clinical population (74%), with Axis 1 DSM IV (Diagnostic and Statistical Manual of Mental Disorders fourth edition) disorders, anxiety disorders, and mood disorders [21].

Binge eating (BE) refers to the uncontrollable eating of large amounts of food in a short period of time and occurs in response to excessive dietary restriction [22], or as a compensatory strategy for coping with anxiety or negative feelings [23], reflecting maladaptive emotion regulation, such as avoidant coping. Numerous studies have shown a relationship between EE and BE, in both clinical and non-clinical samples [24,25], while a significant positive association between BE and body mass index (BMI) has consistently been observed. The relationship between BE and BMI occurs in both clinical and non-clinical populations [26,27,28,29]. The body mass index (BMI) is a mathematical ratio that relates a person’s weight to their height and, according to the gender- and age-specific cut-offs of the International Obesity Task Force (IOTF) standards, is used to determine whether they fall into the normal, underweight, and overweight/obese groups. BMI is expected to be higher in men than in women and to increase with age [30]. In recent decades, childhood obesity has been progressively rising and severe childhood obesity has become a significant public health issue on a national and international level. Lockdown caused by the coronavirus disease 2019 (COVID-19) pandemic is currently causing worry since it may worsen the spread of childhood obesity and widen the risk gap for obesity [31].

The association between perfectionism and BE is questionable: Tachikawa et al., 2004 found that individuals with anorexia who binged had higher perfectionism levels than controls [32], yet according to findings from other researchers, BE is not linked to higher degrees of perfectionism [33]. Other authors, however, have found that the association between BE and perfectionism is weak but significant, suggesting that the relationship between these two variables may be due to the presence of other eating disorders [34]. Perfectionism has been proposed as a risk factor for BE symptomatology [35].

Perfectionism’s significance in the relationships between EE, BE, and BMI may help us better grasp the general risk factors for developing eating disorders. In this context, an empirical study is proposed with the following objectives:To confirm the association between BMI, symptoms of EE, BE, and perfectionism.To determine the factors that influence BMI and symptoms of BE.To investigate how perfectionism affects both EE and BMI.

The following hypothesis are proposed:

**H1:** *Perfectionism will have a direct correlation with EE, BE and BMI*.

**H2:** *Perfectionism will predict BE symptoms and BMI*.

**H3:** *Perfectionism will have a mediator role between EE and BMI*.

### Method

#### Participants and Procedure

A cross-sectional survey was conducted with 312 adult participants, of whom 190 were women (60.9%) and 122 were men (39.1%). The sample was selected by non-probabilistic, convenience and snowball sampling. Exclusion criteria were drug consumption (except smoking or sporadic alcohol intake) and the diagnosis of any physical or psychological problems. The sociodemographic characteristics of the sample are shown in Table 1.

The survey was applied using the Qualtrics platform and was distributed through social media by snowball sampling. The study was approved on 13th June 2019 by the University Francisco de Vitoria Ethics Committee and was conducted in compliance with the Declaration of Helsinki. Prior to completing the survey, participants were given an informed consent form. There was no payment for participation, which was voluntary. The data was completely anonymized.

## 2. Materials and Methods

Participants completed an ad hoc questionnaire with sociodemographic information. To establish the exclusion criteria, participants were asked about their health status (including the Body Mass Index), eating habits, physical activity and drug consumption. The participants then answered to the following questionnaires:

The Emotional Eater Questionnaire (EEQ) [36], used to measure the effect of emotions on eating behavior. The questionnaire consisted of 10 items self-administered with a global score. Items consisted of a Likert scale from 1 (never) to 4 (always). To interpret the global score, the questionnaire allowed cut-offs to be set to determine the emotional eating pattern: non-emotional eater, low-emotional eater, emotional eater, or high-emotional eater. The reliability obtained in validation show acceptable values of internal consistency (Cronbach’s alpha > 0.7) [36,37]. In the present study, the internal consistency was 0.85.

The Multidimensional Perfectionism Scale [38], was used to measure perfectionist behavior. The questionnaire consisted of 35 items with a response format based on a Likert scale from 1 (totally disagree) to 5 (totally agree). The questionnaire can be interpreted by its global score, as well as its 6 dimensions: Concern over Mistakes (CM) (e.g., if I fail at work/school, I am a failure as a person), Personal Standards (PS) (e.g., it is important to me that I am thoroughly competent in everything I do), Parental Expectations (PE) (e.g., my parents set very high standards for me), Parental Criticism (PC) (e.g., as a child, I was punished for doing things less than perfect), Doubts and actions (DA) (e.g., even when I do something very carefully, I often feel that it is not quite right), and Organization (O) (e.g., organization is very important to me). Internal consistency ranged from 0.73 to 0.89 for the dimensions and 0.9 for the global score. In our study, Cronbach’s alpha was 0.88 for CM; 0.83 for PS; 0.83 for PE; 0.71 for PC; 0.78 for DA; and 0.82 for O. For the global score, Cronbach’s alpha was 0.91.

The Binge Eating Scale (BES) [39] is the validated Spanish version [40]. This is a self-administered questionnaire consisting of 16 groups of 3 or 4 sentences. Individuals must choose the sentence that best describes how they feel about problems with eating control. Global scores range from 0 to 46. Each sentence of each group ranges from 0 to 3. Global scores can be classified as absence of binge eating, moderate binges, and severe binge eating. The original Cronbach’s alpha was 0.86 and in the present study the value was 0.89.

To analyze the data, a Pearson correlation coefficient was obtained between all the variables studied. Two multiple regression analyses were then carried out to analyze the predictor variables of BMI and BE disorders. The regression analysis was carried out using forward stepwise variable selection. Finally, the moderation analysis of perfectionist between emotional eating and BMI was calculated. For mediation analysis, the model 4 of mediation were analyzed with a level of confidence of 95% and using 10,000 bootstrap samples. The SPSS version 21 was used for all data analysis and the PROCESS macro v3.5.3 for mediation analysis by Andrew F. Hayes © (https://www.processmacro.org/index.html (accessed on 12 October 2021).

## 3. Results

### 3.1. Correlation Analysis between Variables

Table 2 shows bivariate correlations between the variables studied. EE showed a direct correlation with perfectionism and with all its dimensions except for O (organization) and PE (parental expectations). The correlation between EE symptoms and BMI was also direct and statistically significant. BE symptoms showed a direct significant correlation with the global score of perfectionism as well as CM (concern over mistakes), PC (parent criticism), and DA (doubts and actions) dimensions. Direct and statistically significant correlations were found between BE symptoms and BMI. In contrast, BMI showed an inverse significant correlation with the perfectionism global score, CM, PS (personal standards), and DA dimensions.

### 3.2. Predictive Models of BMI and EE Symptoms

Table 3 shows the four predictive models for BMI. All the models were statistically significant. The most complete model considered the variables age, gender, EE score and DA dimension of perfectionism (F = 27.89; *p* < 0.001), which accounted for 25.7% of the variance of the BMI. Partial correlations showed that the EE score was the most relevant variable in the model (*r*_p_ = 0.343), following by gender (*r*_p_ = 0.309), age (*r*_p_ = 0.305), and DA (*r*_p_ = −0.244).

Regarding the predictive variables of BE symptoms, the results are provided in Table 4. The analysis produced three models. The most relevant of these three models included the EE score as the strongest predictive variable (*r*_p_ = 0.757), followed by the CM and PE dimensions of perfectionism (*r*_p_ = 0.192 and −0.140 respectively). The model accounted for 65% of the variance of BE symptoms.

### 3.3. Mediator Effect of Perfectionism between Emotional Eating and BMI

Figure 1 shows the mediator effect of the global score of perfectionism. As shown, emotional eating has a significant direct effect on perfectionism and BMI. This significant direct effect is also found inversely between perfectionism and BMI. The indirect effect of EE through perfectionism is also significant and inverse.

Regarding the specific dimensions of perfectionism, Figure 2 and Figure 3 show the only two dimensions which had significant indirect effects: CM and DA. In these two models EE showed a significant direct effect on CM and DA, as well as on BMI. However, the significant effects of these dimensions on BMI were inverse, with an indirect effect found in both models.

## 4. Discussion

The present study advances the understanding of the role of personality traits in the relationship between EE and BMI and is an important step toward improving treatments for loss weight or to control EE.

The first aim of this study was to confirm the association between BMI, symptoms of EE, BE, and perfectionism. The first hypothesis is confirmed: perfectionism has a direct correlation with EE, BE, and BMI. Nevertheless, this relationship between BMI, EE, BE, and perfectionism is demonstrated in a non-univocal way: people with a higher overall score in perfectionism show a higher level in EE and BE, but have a lower BMI. However, if we look at the components of perfectionism, there are two dimensions, O (organization) and PE (parental expectations), that are not related to any of the three variables (EE, BE, and BMI). O was associated in the Multidimensional Perfectionism Scale with positive achievement, striving and work habits [38], and is one of the characteristics of healthy perfectionists [41]. Thus, it is not surprising that it is not related to unhealthy eating patterns. The PE subscale, the propensity to think that one’s parents are unduly demanding and critical contributed the smallest variance (3.5%) to the global measure of perfectionism in the creation of the scale [38].

The second aim was to identify the variables that predict BE symptoms and BMI. In our study, BMI was predicted by the EE score and the DA dimension of perfectionism and BE symptoms were predicted by the EE score and the CM and PE dimensions of perfectionism. The second hypothesis is partially confirmed: only two components of perfectionism acted as predictor variables for BE symptoms, and concerning BMI, only a single dimension of perfectionism, DA, was a predictor variable. DA appears to be high in patients with eating disorders [42]. For both BMI and BE symptoms, EE was a more powerful predictor than the dimensions of perfectionism. Regarding BMI, EE, convergently, has been associated with increased BMI in numerous studies [43,44,45], and is one of the factors associated with BE [46]. A recent systematic review of the relationship between EE and body weight in adults showed that both longitudinal studies and intervention studies have found associations between EE, BMI and weight gain, or less weight loss success [46]. In the same line, a recent study shows that participants with a higher BMI who reported more EE were more prone to BE (Černelič-Bizjak & Guiné, 2021). Contrarily, CM was also an important predictor of BMI. CM (concern over making mistakes) is the aspect of perfectionism most related to eating disorder psychopathologies, specifically within AN [47], and previous research has found this component of perfectionism to be predictive of BE [48]. According to the WHO, obesity has reached epidemic proportions [49] and the evidence that EE and CM are predictors of increased BMI suggests that these two factors cannot be ignored and should be two of the targets in obesity prevention and weight loss programs. A holistic approach to obesity prevention should encompass the detection of both weight loss and the reduction of uncontrolled eating; the recognition of individuals with EE or BE symptomatology is critical to reducing the high risk of obesity. Furthermore, certain kinds of eating disorders, including binge eating and bulimia nervosa, are high among individuals with excess weight [44], and psychosocial interventions should consider the control of EE and the CM aspect of perfectionism to improve their symptomatology.

The third aim of this study was to examine whether perfectionism or its components mediate between EE and BMI. The third hypothesis is partially confirmed: EE showed significant direct effect on BMI. However, the presence of these two dimensions reverses the relationship between EE and BMI (a higher EE score corresponds to a lower BMI when scores in DA and CM are high). DA and CM are critical components to understanding the association between EE and BMI. 

In recent decades, numerous studies have attempted to explain the relationship between EE and BMI, and most concluded that individuals with high levels of EE have a higher BMI than those with lower levels [36,50,51]. However, our study suggests there is a mediating variable between EE and high BMI: perfectionism. According to evidence, perfectionism should be viewed as a complex and multidimensional construct [6,52]. Specifically, two of the components of this multidimensional construct (DA and CM) have a mediating role in the relationship between EE and BMI. In developing the Frost scale [38], the principal dimension was excessive CM (concern over making mistakes); thus, it is not surprising this component is a significant mediating variable compared to the others. The study by Frost & Marten also found that CM and DA (doubting of the quality of one’s actions) were more closely related to psychopathology than other dimensions such as PS (personal standards) and O (organization).

DA and CM seem to interact with EE to produce two different (pathological) eating patterns: high scores in these two components in those with EE is related to intake restriction (and low BMI), while low scores are related to overeating (and consistently, high BMI). These findings have strong ramifications for our comprehension of how eating disorders develop, and confirm the relationship between anorexia or restriction of food intake and perfectionism found in previous EE research [53,54]. Studies have found evidence that it is possible to adjust perfectionism in non-clinical samples [55], and consequently to change eating behaviors and attitudes [56].

Among the future developments of this line of research, it is worth mentioning the comparative analysis of gender and different age groups. These variables, according to regression models, predict BMI. The goal of the current study was to investigate the association between perfectionism, EE, BE symptoms, and BMI among the general population. The next set of objectives of our research includes examining differences based on gender and age.

There are certain limitations on this study that affect how the findings should be interpreted. First, only one perfectionism measure was applied. Second, although a mediation model was tested in which the directions of effects were suggested, we should be careful in interpreting these findings. Given that recent research has shown there is not only significant between-person variability but also within-person variability in perfectionism [57], and thus more dynamic and bidirectional relations may take place. Third, a physical or psychological problem’s diagnosis was one of the exclusion criteria. This exclusion criteria seems general and is based solely on the participant’s own evaluation without any official assessment, which is another study limitation. Finally, investigating the relationships among higher-order personality traits, perfectionism, and eating habits would be useful.

## 5. Conclusions

Perfectionism has a direct correlation with EE, BE, and BMI. EE was the most powerful predictor of BMI and BE symptomatology. Further, BMI was also predicted by the DA dimension of perfectionism and BE symptoms were also predicted by the CM and PE dimensions of perfectionism. The mediating effect of two dimensions of perfectionism (DA and CM) on EE is confirmed: a higher EE score corresponds to a lower BMI when scores in DA and CM are high. The various dimensions of perfectionism must be considered when designing and implementing programs for the prevention and intervention in obesity and eating pathologies.

## Figures and Tables

**Figure 1 nutrients-14-03361-f001:**
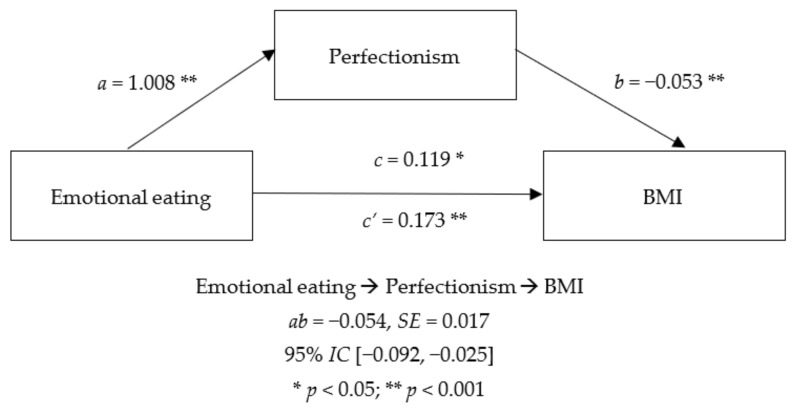
Mediator effect of perfectionism between emotional eating and BMI.

**Figure 2 nutrients-14-03361-f002:**
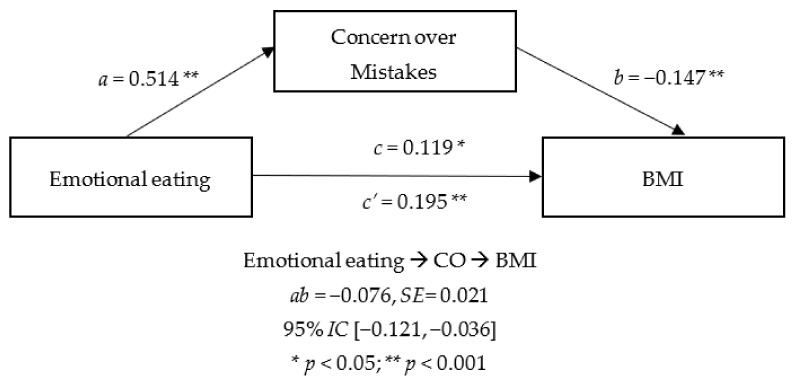
Mediator effect of CM between emotional eating and BMI.

**Figure 3 nutrients-14-03361-f003:**
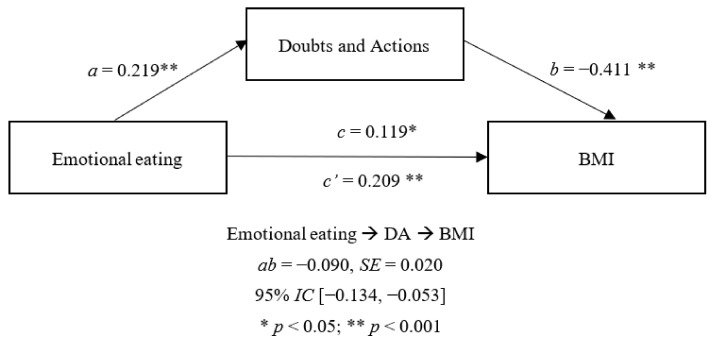
Mediator effect of DA between emotional eating and BMI.

**Table 1 nutrients-14-03361-t001:** Sociodemographic characteristics of the sample.

	*n* = 212
Mean age	32.04 (SD = 11.86)
Education	
Without studies	4 (1.3%)
Primary studies	26 (8.3%)
Secondary studies	88 (28.2%)
University studies	194 (62.2%)
Marital status	
Single	137 (43.9%)
With partner	97 (31.1%)
Married	69 (22.1%)
Separated	9 (2.9%)
Employment status	
Student	84 (26.9%)
Unemployed	42 (13.5%)
Self-employed	33 (10.6%)
Salaried	153 (49.0%)

**Table 2 nutrients-14-03361-t002:** Bivariate correlations between the variables.

	1	2	3	4	5	6	7	8	9
1. Emotional Eater	-								
2. Perfectionism	0.297 **								
3. CM	0.411 *	0.836 **							
4. PS	0.113 *	0.749 **	0.519 **						
5. PE	0.098	0.676 **	0.395 **	0.37 **					
6. PC	0.26 **	0.706 **	0.553 **	0.33 **	0.639 **				
7. DA	0.355 **	0.72 **	0.642 **	0.419 **	0.329 **	0.485 **			
8. O	−0.091	0.37 **	0.09	0.267**	0.165**	0.031	0.122 *		
9. Binge Eating	0.799 **	0.269 **	0.411 **	0.095	0.036	0.214 **	0.35 **	−0.092	
10. BMI	0.163 **	−0.179 **	−0.143 *	−0.130 *	−0.084	−0.043	−0.246 **	−0.092	0.169 **

CM: Concern over Mistakes; PS: Personal Standards; PE: Parental Expectation; PC: Parental Criticism; DA: Doubts and Actions; O: Organization; BMI: Body Mass Index. * *p* < 0.05; ** *p* < 0.001.

**Table 3 nutrients-14-03361-t003:** Predictive models of BMI.

Model	Variables	β	t	R^2^	ΔR	Change in F
1	Age	0.271	4.96 **	0.074	0.071 **	24.64
2	Age	0.297	5.61 **	0.146	0.14 **	26.01
	Gender	0.269	5.1 **			
3	Age	0.325	6.33 **	0.207	0.199 **	23.77
	Gender	0.32	6.15 **			
	Emotional eater	0.254	4.87 **			
4	Age	0.281	5.6 **	0.267	0.257 **	25.00
	Gender	0.287	5.68 **			
	Emotional eater	0.338	6.39 **			
	DA	−0.267	−5.00 **			

DA: Doubts and Actions. ** *p* < 0.001.

**Table 4 nutrients-14-03361-t004:** Predictive models of BE symptoms.

Model	Variables	β	t	R^2^	ΔR	Change in F
1	Emotional Eater	0.799	23.38 **	0.638	0.637 **	546.64
2	Emotional Eater	0.758	20.42 **	0.646	0.644 **	7.098
	CM	0.099	2.66 *			
3	Emotional Eater	0.751	20.33 **	0.653	0.65 **	6.13
	CM	0.138	3.44 *			
	PE	−0.091	0.014 *			

EEQ: Emotional Eater Questionnaire; CM: Concern over Mistakes; PE: Parental Expectation. * *p* < 0.05; ** *p* < 0.001.

## Data Availability

The data described in this manuscript are not available for reasons of confidentiality. Nevertheless, the database anonymized will be made available upon request pending.

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
