# Peer review of "Emotional Eating and Perfectionism as Predictors of Symptoms of Binge Eating Disorder: The Role of Perfectionism as a Mediator between Emotional Eating and Body Mass Index"

_nutrients, 2022, doi:10.3390/nu14163361_

Round 1

Reviewer 1 Report

This study has potential but I was confused about your aims and conclusions throughout my experience of reading the paper and was required to continually flick back and forth to try and make sense of it. 

The title indicates that this study is about the potential predictive roles of perfectionism and emotional eating on binge eating and yet you didn't explore the mediating role of perfectionism with regard to predictive variables for binge eating. 

Line 142-143 "EE showed a direct correlation with perfectionism and with all its dimensions except for O (organization)." According to Table 2 this statement is incorrect as EE also did not have a significant correlation with PE.

Line 143-144 & 147-148 " Direct and statistically significant correlations were found between BE symptoms and BMI." and "The correlation between BE symptoms and BMI was also direct and statistically significant." These both state the same thing ??

Table 3 - It is not clear to me why you have built your model based on Age and Gender as the initial predictive variables (and why binge eating is not one of the variable you are looking into as a predictor of BMI?). The rationale for the model is not strongly enough identified in your introduction section nor in your procedure/analysis section. Therefore the selection of these variables for your model is unclear. Please provide more detail on your reason for this design. 

As noted before, I am unclear as to why you were only interested in the mediation of perfectionism on BMI and not the mediating role of perfectionism on predictors of binge eating.

Writing clarity needs improvement throughout to better communication your aims, design and the interpretations of the results of the study. 

Author Response

Dear reviewer.

Thank you very much for the reviews and suggestions that have helped improve our article. We attach the responses to your reviews. We are at your disposal for any matter.

Sincerely.

----

This study has potential but I was confused about your aims and conclusions throughout my experience of reading the paper and was required to continually flick back and forth to try and make sense of it.

The title indicates that this study is about the potential predictive roles of perfectionism and emotional eating on binge eating and yet you didn't explore the mediating role of perfectionism with regard to predictive variables for binge eating.

We assume a multidimensional conception of perfectionism. From line 170 the regression analysis is described where the dimensions of perfectionism with predictive value are reflected (Table 4). However, we have clarified the wording of this part.

Line 142-143 "EE showed a direct correlation with perfectionism and with all its dimensions except for O (organization)." According to Table 2 this statement is incorrect as EE also did not have a significant correlation with PE.

You are right, thanks for the warning, we have modified the text that explains table 2

Line 143-144 & 147-148 " Direct and statistically significant correlations were found between BE symptoms and BMI." and "The correlation between BE symptoms and BMI was also direct and statistically significant." These both state the same thing ??

Thanks. The second sentence should say “The correlation between EE symptoms and BMI was also direct and statistically significant”. We have modified the entire paragraph by organizing the description of the correlations (152-155).

Table 3 - It is not clear to me why you have built your model based on Age and Gender as the initial predictive variables (and why binge eating is not one of the variable you are looking into as a predictor of BMI?). The rationale for the model is not strongly enough identified in your introduction section nor in your procedure/analysis section. Therefore the selection of these variables for your model is unclear. Please provide more detail on your reason for this design.

The BE variable was not significant in the regression analysis, and for this reason it was not included in the results table. And we have added the justification for the inclusion of the gender and age variables in the introduction (61-65). The fact that these variables, as expected, are predictors, supports the relevance of the model.

As noted before, I am unclear as to why you were only interested in the mediation of perfectionism on BMI and not the mediating role of perfectionism on predictors of binge eating.

The body mass index (BMI) is a mathematical ratio that is used to assess whether they are overweight or obese. We think it's crucial to identify the determinants of this variable given the significance of obesity for population health overall and the rise in overweight in first-world cultures. So, we can develop better strategies for improving the health of both populations and individuals. We have added a paragraph about this in the introduction (65-70).

Writing clarity needs improvement throughout to better communication your aims, design and the interpretations of the results of the study.

To make the topic clearer, we changed the wording in the highlighted portions.

Reviewer 2 Report

The manuscript is on an interesting issue, namely the potential role of perfectionism as mediator between Emotional Eating, Binge Eating, and BMI, in a non-clinical sample. The sample size is adequate, as well as the statistical method applied. Results are clearly described and discussed. According to study results, perfectionism was directly correlated with Emotional Eating, Binge Eating and BMI, and was a predictor of BMI increase. 

Here some shortcoming to report:

1) The sample has been analyzed as a whole. I was wandering if there were differences between males and females, considering that males constituted around 39% of the sample.

2) It could be interesting to know if there were differences between younger vs. older participants, and if there was any inclusion/exclusion criterion based on age.

3) One of the exclusion criteria was 'the diagnosis of any physical or psychological problem'. This exclusion criterion is based only on a self evaluation by the participant, without any formal assessment, and sounds generic. This is a study limitation.

Author Response

Dear reviewer.

Thank you very much for the reviews and suggestions that have helped improve our article. We attach the responses to your reviews. We are at your disposal for any matter.

Sincerely.

--

The manuscript is on an interesting issue, namely the potential role of perfectionism as mediator between Emotional Eating, Binge Eating, and BMI, in a non-clinical sample. The sample size is adequate, as well as the statistical method applied. Results are clearly described and discussed. According to study results, perfectionism was directly correlated with Emotional Eating, Binge Eating and BMI, and was a predictor of BMI increase.

Here some shortcoming to report:

1) The sample has been analyzed as a whole. I was wandering if there were differences between males and females, considering that males constituted around 39% of the sample.

2) It could be interesting to know if there were differences between younger vs. older participants, and if there was any inclusion/exclusion criterion based on age.

1 & 2) We appreciate your remarks very much.

Gender and age, according to regression models, predict BMI. The goal of the current study was to investigate the association between perfectionism, EE, BE symptoms, and BMI among the general population. Due to this, we have omitted the analyses of variations across genders and age groups. The next set of objectives of our research includes examining differences based on gender and sex.

We have included this purpose as prospective at the end of the manuscript (before the limitations of the study) (263-268).

3) One of the exclusion criteria was 'the diagnosis of any physical or psychological problem'. This exclusion criterion is based only on a self evaluation by the participant, without any formal assessment, and sounds generic. This is a study limitation.

Thanks for the warning, we have added this limitation to our study
